# A Critical Review on the Properties and Applications of Sulfur-Based Concrete

**DOI:** 10.3390/ma13214712

**Published:** 2020-10-22

**Authors:** Roman Fediuk, Y. H. Mugahed Amran, Mohammad Ali Mosaberpanah, Aamar Danish, Mohamed El-Zeadani, Sergey V. Klyuev, Nikolai Vatin

**Affiliations:** 1Department of Hydraulic Engineering and the Theory of Constructions, Far Eastern Federal University, 690091 Vladivostok, Russia; 2Department of Civil Engineering, College of Engineering, Prince Sattam Bin Abdulaziz University, Alkharj 11942, Saudi Arabia; m.amran@psau.edu.sa; 3Department of Civil Engineering, Faculty of Engineering and IT, Amran University, Amran, Quhal 9677, Yemen; 4Civil Engineering Department, Cyprus International University, 99258 Nicosia, North Cyprus, Turkey; mmosaberpanah@ciu.edu.tr (M.A.M.); aamardanish@gmail.com (A.D.); 5Department of Civil Engineering, Universiti Putra Malaysia, Serdang 43400, Malaysia; mohamed.elzeadani9595@gmail.com; 6Department of Strength of Materials and Structural Mechanics, Belgorod State Technological University n.a. Shukhov, 308012 Belgorod, Russia; klyuyev@yandex.ru; 7Higher School of Industrial, Civil and Road Construction, Peter the Great St. Petersburg Polytechnic University, 195251 St. Petersburg, Russia; vatin@mail.ru

**Keywords:** sulfur, binder, modifier, sulfur-based concrete, melt, fresh properties, hardened properties

## Abstract

The incessant demand for concrete is predicted to increase due to the fast construction developments worldwide. This demand requires a huge volume of cement production that could cause an ecological issue such as increasing the rates of CO_2_ emissions in the atmosphere. This motivated several scholars to search for various alternatives for cement and one of such alternatives is called sulfur-based concrete. This concrete composite contributes to reduce the amount of cement required to make conventional concrete. Sulfur can be used as a partial-alternate binder to Ordinary Portland Cement (OPC) to produce sulfur-based concrete, which is a composite matrix of construction materials collected mostly from aggregates and sulfur. Sulfur modified concrete outperforms conventional concrete in terms of rapid gain of early strength, low shrinkage, low thermal conductivity, high durability resistance and excellent adhesion. On the basis of mentioned superior characteristics of sulfur-based concrete, it can be applied as a leading construction material for underground utility systems, dams and offshore structures. Therefore, this study reviews the sources, emissions from construction enterprises and compositions of sulfur; describes the production techniques and properties of sulfur; and highlights related literature to generate comprehensive insights into the potential applications of sulfur-based concrete in the construction industry today.

## 1. Introduction

Sulfur is a by-product of large-tonnage waste from oil and gas production facilities. The forecast for the global production of sulfur (based on data from the US Geological Survey (USGS) [1]) makes it important to seriously consider various alternative uses of sulfur as well as its complete utilization (disposal). In recent years, China has been the biggest contributor of sulfur as a by-product, due to increase in the number of refineries and gas processing plants (17 million tons in 2018) [2]. After China, the largest producers of sulfur are the USA (9.7 million tons), Russia (7.1 million tons), Saudi Arabia (6 million tons), Canada (5.5 million tons), Japan and Kazakhstan (both = 3.5 million tons each) with total annual production of 80 million tons in 2018 [2]. One of the promising ways to utilize sulfur is in construction as a constituent of sulfur-based concrete. Advantages of sulfur-based concrete as compared to Portland cement concrete are; rapid curing [3], waste management [4], possibility of recycling [5], high resistance to acids and radiation [6], possibility of concreting at negative ambient temperatures [7], quick setting time [8], low electrical and thermal conductivity [9], water tightness [10], high frost resistance [11] and high wear resistance [12] as shown in Figure 1.

Sulfur has a large number of different allotropic modifications due to the high ability of its atoms to combine with each other to form ring or chain molecules. Allotropes of sulfur can be classified into two types of intramolecular allotrope (formed due to chemical bonding between sulfur atoms) and intermolecular allotrope (formed due to sulfur molecule’s arrangement within crystals) [13]. Sulfur atoms unite to form chains (catena/polycatena sulfur) and cyclic rings (cyclo-S_n_: where n represents number of atoms) which allows millions of sulfur allotropes (intramolecular) to exist (considering all sulfur atoms’ possible combinations). It should be noted that if S_n_ molecules have 6–12 sulfur atoms, they exist in the form of rings (very stable) while molecules with less than six or greater than twelve sulfur atoms can either exist in the form of ring or chain (unstable).

Allotropes differ in physical properties and are quite similar in most chemical properties [13]. They can exist together in equilibrium in certain proportions, depending on temperature and pressure. The presence and concentration of each allotrope and, consequently, the physical and chemical properties of solid sulfur depend on the thermal history. The most important allotropic modifications are rhombic (α), monoclinic (β) [15] and plastic. Sulfur α forms rhombic crystal and is stable below 95.5 °C (melting point 112.8 °C); sulfur β forms monoclinic needles and is stable between 95.5 °C and melting point (119.3 °C). Both are made up of cyclo-S_8_ molecules. Liquid sulfur above 159 °C is a solution of linear chains, without regular arrangement between them, formed by opening S_8_ rings and polymerizing them [3].

Further, several studies focused on sulfur-based concrete, initially in North America since the 1970s [16,17,18,19,20,21]. Based on the major findings of previous research, sulfur-based concrete was seen to be sufficiently safe for the environment. In the period from 1980s–1990s, the upsurge in hydrocarbon production initiated greater sulfur retrieval as a by-product of gas and oil [22,23,24]. Sulfur can be used as a partial-alternate binder to OPC to produce sulfur-based concrete. Sulfur-based concrete is a composite matrix of construction materials made up mostly of aggregate and sulfur. Due to several superior characteristics that sulfur-based concrete has over traditional concrete as given in Table 1, it is highly recommended for use in underground utility systems, dams and offshore structures. Therefore, this study aims to present a state-of-the-art review on sulfur-based concrete to provide comprehensive insights into the potential applications of sulfur in the construction industry today. This includes first a review on the natural and man-made sources of sulfur, followed by the emissions from construction enterprises and compositions of sulfur. After that, sulfur production techniques and properties are thoroughly discussed, followed by the applications of sulfur-based concrete and directions for future research.

## 2. Source of Sulfur

Sulfur can be obtained from both natural and man-made sources; although, much of the sulfur obtained worldwide from both sources is very difficult to quantify. For instance, sulfur obtained from mining and byproducts of environments (like oil refineries, processing plants of natural gas and smelters of nonferrous metals) can be reasonably quantified. However, sulfur obtained from industries and electric power plants are very difficult to define. Additionally, sulfur emissions obtained through natural sources are difficult to quantify because of the variability of sources, emissions and compounds involved [1].

### 2.1. Natural Sources of Sulfur

The natural sources of sulfur production are very complex and difficult to quantify. Sulfur is available in various minerals in the crust of earth, which makes it one of the very few elements to be present in earth’s crust in elemental state. It is also available in coal, oil and natural gas in the form of various compounds and quantities. Additionally, sulfur is an essential part of all living creatures, including plants and animals [1,29]. Minerals of sulfide available in lithosphere are weathered to produce sulfates, out of which some is discharged into oceans by various processes like erosion and river runoff. The leftover weathered sulfate produces several compounds by reactions with bacteria, which are eventually incorporated into plant/soil systems. Animals use/eat plants containing sulfur constituents, which eventually produce sulfates after the ingestion process [29]. Volcanoes are the most dramatic but best known naturally available sources of sulfur. The sulfur compounds emitted during volcanic activities happen during non-eruptive time as well as during eruption. Moreover, seawater also contains sulfur in a way that every gram of water contains 2.56 mL of sulfate. The availability of sulfur in water bodies is due to weathered minerals and decay of underwater species. As the bubbles (molecules) of water, in particular, at sea, river, ocean or any water body break, salt particles are formed and get into atmosphere. Naturally available sources of sulfur contribution are shown is Figure 2.

### 2.2. Man-Made Sources of Sulfur

Although, the quantity of sulfur emitted into atmosphere and sulfur cycle due to human activities is very easy to quantify as compared to natural sources [30], the majority of man-made sulfur resources are due to fossil fuel (like coal, natural gas and petroleum) burning, smelting of ores (nonferrous metals) and various industrial/burning processes [31,32,33,34]. Man-made sulfur emissions into atmosphere began to increase extensively during the 20th century. The increasing trend continued until the 1970s, after which environmental regulations on sulfur emissions were imposed in America and Europe [35]. Although environmental regulations decreased sulfur emissions; however, it did not eliminate the problems completely.

#### 2.2.1. Natural Gas

The recovery of sulfur from natural gas starts with H_2_S separation. The separation of H_2_S is necessary due to its toxicity and corrosive nature. The natural gas constituting H_2_S is called sour gas which is made to pass from a solvent (like amines) [37,38] in which H_2_S dissolves and the required percentage of natural gas remains insoluble. Then, the solvent is heated causing H_2_S to be removed from the solution [37,38]. After the separation of several component of natural gas, H_2_S is then converted into sulfur by processes like Claus [1]. This method produces elemental sulfur as a by-product material.

#### 2.2.2. Petroleum

Crude oil generally constitutes carbon (84%), hydrogen (14%), sulfur (1–3%) and nitrogen/oxygen/metals/salt (<1%) [39]. The petroleum refining process causes sulfur to separate from the various organic compounds in the form of H_2_S. Until the environmental regulations of the 1970s, H_2_S was used as refining fuel which was obtained during the refining process. This process was limited as burned H_2_S releases sulfur dioxide into atmosphere. In general, The H_2_S produced in oil/petroleum refinery can be further processed to produce elemental sulfur [40].

#### 2.2.3. Oil Sands

One of the problems of oil sand reserves development is a rise of sulfur and nitrogen deposition in that region [41]. Mainly this oil is intensely buried and steam injection is needed which causes a variety of emissions including H_2_S, CO_2_ and H_2_ [41,42]. Sands contribute to a significant source of sulfur, which are predominantly found in Canada where approximately 300 billion barrels of extractable oil having 3.5–5% sulfur [43,44]. Generally, oil sands are a combined form of bitumen, clay, water and sand. Oil sands having 10% and 7% bitumen are considered to be rich and not cost effective [43]. The improvement of oil sand refinery must be done to obtain a significant amount of sulfur. In particular, it could be said that the production of H_2_S is under special attention as on the one hand, it plays a role in oil partial desulphurization and on the other hand, due to its toxicity, there is some difficulty to handle and transport it [42].

#### 2.2.4. Sulfide Smelting

Combined sulfur can be obtained through nonferrous metal smelting. Smelter gases having SO_2_ is transformed into H_2_SO_4_ and liquid sulfur. In the US (1990), approximately 11% of sulfur was produced through H_2_SO_4_ obtained from smelting nonferrous metals [45]. Sulfur can also be obtained from SO_2_ emissions. In order to obtain useful sulfur or to dispose it, the desulfurization process can be applied which uses compounds like CaO, NaCO_3_, MgO to get SO_2_ while it produces elemental sulfur, CaSO_4_.2H_2_O, liquid sulfur and H_2_SO_4_ [36]. Figure 3 shows a schematic shape of flash smelting furnace where the slag component is mixed with sulfate mineral concentrate which will be injected into the furnace with enriched oxygen until oxidation reaction shapes [46]. Thereafter, the molten matte and slag which are heavier will be separated and fall down to the bottom of the furnace [46]. This process makes SO_2_ and other harmful emissions. Throughout the process, the gas given off is converted to sulfuric acid, which can be used as a by-product [47].

## 3. Production of Sulfur

Natural reserves of sulfur (including sulfur ores of sedimentary and magmatic genesis) amount to more than 5 billion tons. Of these, explored deposits of native sulfur have a capacity of about 1.2 billion tons. The mining industry of sulfur is divided into two sectors: specialized and attendant. The specialized part is mainly aimed at the extraction of sulfur from the deposits of this raw material (one tenth of the total sulfur production on the planet). The specialized sulfur native ores are in Iraq (about 335 million tons) [48], the United States (200 million tons) [49], Chile (100 million tons) [50] and Mexico (100 million tons) [51]. Large deposits have also been explored in Poland [52], Ukraine [53], Russia [54], Turkmenistan [55] and on the Japanese islands [56]. In the attendant sector, sulfur is produced as by-products in the process of hydrogen sulfide processing; the level of sulfur production depends not only on the volumes of its consumption but on the amount of purified oil and natural gas. Commercial production of sulfur has three types such as lump, granulated and liquid. Sulfur production technologies includes extraction and refining of natural elemental sulfur [57], obtaining sulfur from pyrites [58], sulfur production from H_2_S [59,60] and sulfur production from SO_2_ [61].

### 3.1. Claus Process

As every source of sulfur (especially man-made) produces H_2_S which can be converted into sulfur by several processes worldwide. One of such methods is known is Claus (named after its inventor Carl Friedrich Claus) [30,62]. The overall efficiency of the Claus process is between 94–97% [63,64]. The traditional Claus process (shown in Figure 4) is carried out as follows [64]

H_2_S and the oxygen (O_2_) available in the air are reacted to form SO_2_

The above reaction produces a lot of heat while H_2_S and SO_2_ reacts with each other to produce 3/2 S_2_. This reaction is high reversible exothermic reaction which lessens equilibrium transformation up to 75%. Effluent gas produced in reaction furnace is transferred to waste heat boiler (WHB) to reobtain heat and form steam (high-pressure). The S_2_ present in effluent gas changes to hexasulfur (S_6_) and octasulfur (S_8_)

WHB having effluent gas is transferred to a condenser for condensing sulfur. Condensed effluent gas is heated and transferred to 2–3 catalytic reactors. Each catalytic reaction stage produces sulfur due to cooling converted effluent gas in condenser.

### 3.2. Frasch Mining

Dr. Herman Frasch in 1984 invented a process to recover sulfur by melting it underground and then pumping it upward to the surface [1]. This process was first used commercially at Sulfur Mine, LA in 1903 [65]. Frasch mining (as shown in Figure 5) usually proceeds in the following steps [37]:

Frasch pump is inserted in the ground containing sulfur deposits.

Hot water (165 °C) is introduced into mineral strata containing sulfur.

Hot water melts sulfur which is pumped to the surface by pressured air.

Although about 90% of man-made sources of sulfur were recovered from Frasch mining, especially in sulfur rich countries like the US, Iraq and Mexico, this mining method requires the following conditions [66]:Huge, porous and rich sulfur deposits.Impermeable covering rock above the deposit.Reliable and sufficient water supply.Cost effective source of fuel required to heat large water quantities needed to melt the sulfur deposit and to provide enough power required for proper functioning of energy-consuming machineries of the process.

Geological conditions should be satisfied which mean that the deposits should be either bedded evaporite and salt domes having permeable sulfur (packed in impermeable formations) [67].

Although this process is extensively being used for the recovery of sulfur; however, it is not applicable to small and shallow sulfur deposits.

## 4. Properties of Sulfur

### 4.1. Melting/Freezing Point

Sulfur has several melting/freezing points that are commonly dependent upon solid allotrope under consideration (melted) which is shown in Table 2. Decrement in freezing point of sulfur occurs naturally due to the dissociation (automatic) of the melt to produce sulfur, using various solid allotropes, having lower freezing point as compared to cyclo-S_8_ [68]. Hence, a whole mixture has a lower freezing point accordingly. The maximum intensity/concentration of sulfur can be achieved at a known temperature which represents the low freezing point and is known as the natural melting point. The freezing point of sulfur depends on the temperature and pressure of the mixture/melt [13,69,70].

### 4.2. Viscosity

The viscosity of sulfur depends to a large extent on the temperature. For instance, at 160 °C the viscosity of sulfur decreases by up to 7–8 centipoise, after which viscosity of sulfur increases significantly (approximately 930 poise) at 190 °C and then plummets again. The increase/decrease in viscosity also depends on the concentration/intensity and total length of sulfur chains in the liquid. In view of this, decrement in viscosity (at 160 °C) can be attributed to increase in concentration/intensity and total length of sulfur chains while the decrement in viscosity (after 190 °C) can be justified by the decrease in total length of sulfur chains.

### 4.3. Density

Like viscosity, density of sulfur also depends on the temperature. The density of sulfur increases with decrease in temperature as shown in Figure 6. It is reported that as the temperature increases, the polymerization form will be changed from 8 membered rings of sulfur atoms to a long chain with around 10^6^ million atoms which this new polymerization shape reduces the density of sulfur [77]. However, there is a constant temperature at which the polymerization of sulfur changes its several properties (like viscosity and density). This temperature is known as Lambda Temperature [69]. Sulfur is known as an element which has the largest number of solid allotropes and most of them have cyclic molecules with ring size range between 6 and 10 [78]. The density of various allotropes of sulfur in Lambda Temperature is given in the Figure 7.

### 4.4. Color

Different allotropes and melts of sulfur have different colors [87] as shown in Table 3. For instance, pure sulfur at its melting point has a clear and a bright yellow color which continuously changes to deep/opaque red at its boiling point [88]. As sulfur is recovered in the molten/melt state, the cooling rate plays an important role in defining the color of sulfur [89]. For example, if the molten sulfur is cooled at the temperature −80 °C (boiling point) the color would be yellow; however, if the melt is cooled at temperature −209 °C (in liquid nitrogen), red colored sulfur will be obtained [69].

### 4.5. Thermal Conductivity

Like density and viscosity, thermal properties of sulfur also suffer from discontinuity due to polymerization at Lambda Temperature [69]. There is a linear relation between thermal conductivity and temperature. First, by increasing the temperature, the thermal conductivity of sulfur decreases till it reaches to the phase change from solid (monoclinic) to liquid sulfur; thereafter, after a fall, the thermal conductivity rises by increasing the temperature [90]. Thermal conductivity of sulfur is less than most of the rocks and nearly equal to insulative materials like mica/asbestos. Moreover, thermal conductivity of sulfur (solid/liquid) at respective atmospheric pressure is dependent on temperature [91]. It is concluded that thermal conductivity of solid sulfur is greater than liquid sulfur [91].

### 4.6. Strength

Sulfur’s strength depends on its thermal history and purity. Two researchers (Dale and Ludwig) comprehensively studied the compressive and tensile strength of sulfur in 1965 [25]. It is reported that the compressive strength of sulfur ranges from 1800 psi to 3300 psi (12.41 MPa to 22.75 MPa) while the tensile strength depends largely on thermal histories, cooling rate and temperature [6,26,92]. For instance, rapid cooling rate and high initial temperature of molten sulfur produces high tensile strength as shown in Figure 8. Additionally, the strength (compressive, flexural and tensile) of sulfur can be enhanced by modifying its properties through the use of some filler (same phenomenon as cement). For instance, a study revealed that using optimum quantity of filler (limestone) between 15–20% can enhance compressive, tensile and flexural strength to 2.5, 5 and 7 times (without using sulfur) respectively [26].

## 5. Sulfur in the Concrete Industry

Sulfur has been utilized in several industries (like agriculture, petroleum and pharmaceutical industries) and due to the increasing environmental concern of cement production and the depletion material resources for cement ingredients, sulfur has become a valuable binding material as shown in Figure 9 [93]. Additionally, sulfur has also been used to manufacture bitumen which is an essential material to construct roads. sulfur-based concrete can be used to make roadblocks and sidewalks, drainage/sewerage pipes, foundation coverage, railway ties, bridge decks and acid tanks [50,94]. Meanwhile, sulfur asphalt can be used to construct highways, roads and streets. sulfur-based concrete is becoming more popular due to its properties (like higher strength, impermeability, rapid strength development, corrosion resistance and recyclability), which could make it a reliable substitute for cementitious materials. Table 4 shows a detailed comparison of sulfur-based concrete and traditional concrete.

### 5.1. Composition and Mixing of Sulfur-based Concrete

Sulfur-based concrete uses sulfur as a binder in its molten state [7], which replaces ingredients of conventional concrete like water and cement. Sulfur is heated to make molten sulfur, which is cooled to form hardened concrete [92]. The mixing procedure of sulfur-based concrete is performed differently to conventional concrete. Extra care is provided while mixing sulfur-based concrete due to the following reasons [96]:To enhance acid and salt resistance and reduce moisture absorption.To maintain (enhance) mechanical properties of sulfur-based concrete as per conventional concrete, maintain workability and minimize drying shrinkage after hardening.

### 5.2. Sulfur Emissions from Construction Enterprises

Sulfur dioxide emissions during the production of cement are primarily related to the content of volatile or active sulfur in the raw materials and, to a lesser extent, the quality of the fuel used for energy [32,97]. In particular, raw materials with a high content of organic sulfur or pyrite (FeS) lead to increased emissions of sulfur dioxide. According to the National Bureau of Statistics of China, the total emissions of SO_2_ in this country alone in 2015 amounted to 18.6 million tons and the contribution of the cement industry (the third largest source of sulfur dioxide) was 1.47 million tons of SO_2_ (about 7–8%) [27,50]. To reduce emissions of sulfur waste into the atmosphere, in addition to the standard implementation of the technology of heat treatment of raw materials, it is recommended to use the following methods to control air pollution by reducing emissions of sulfur dioxide:The use of vertical grinding units and the passage of waste gases through the mill for heat recovery and to reduce the sulfur content in the gas. In a mill, a gas containing SO_2_ is mixed with calcium carbonate from raw materials and forms calcium sulfate (gypsum) [97].Selection of low sulfur fuel [98] and the introduction of adsorbents, such as hydrated lime, calcium oxide or fly ash with a high content of CaO, into the exhaust gases to the filters [99].The use of wet or dry scrubbers. Dry gas cleaning is more expensive, so this method is used less frequently than wet gas cleaning and is usually used when sulfur dioxide emissions can exceed 1500 mg/m^3^ [100].Emissions of sulfur dioxide in the production of lime are usually lower than in the production of cement, due to the lower sulfur content in raw materials. The followings are recommended ways to reduce emissions of sulfur dioxide:Selection of low volatile quarry materials [101].Introduction of hydrated lime or bicarbonate into the waste gas stream before the filters [102] and injection of highly dispersed quicklime or slaked lime into the cap of the kiln furnace [103].

Moreover, the use of sulfur, which is a waste of enterprises, for the production of building materials is efficient due to waste disposal and contribution to environmental protection. Taking into account the emission into the atmosphere of large quantities of anthropogenic sulfur (170–180 million tons per year) in the composition of flue gases, we are talking about hundreds of millions of tons of “waste” sulfur raw materials [104]. In brief, it could make it as an ideal option to use instead of cement in concrete industry.

### 5.3. Modified Sulfur-based Concrete

Gracia et al. [50] in 2002 put forward the hypothesis that the use of unmodified sulfur is promising to achieve the necessary properties of special concretes. However, this version does not find further continuation. The scientific schools of Korolev [54], Mohamed [5,9,36] and McBee [105] unanimously argue that the use of sulfur without modification at the present stage of development of building materials science seems unlikely. The most common modifiers used to avoid the conversion of sulfur from the monoclinic to the orthorhombic state are dicyclopentadiene or a combination of dicyclopentadiene, cyclopentadiene and diphentene, as well as olefin polysulfide additives [5,106]. However, the limited use of these modifiers in the construction industry is due to the fact that the reaction between dicyclopentadiene and sulfur is exothermic and necessitates cautious temperature control; besides, the sulfur binder modified with dicyclopentadiene is unbalanced when visible to high temperatures [5]. The process of chemical interaction of sulfur with a modifier is similar to the process of sulfur vulcanization of rubbers and proceeds in two stages [107]:

The conversion of sulfur into the reaction forms as a result of thermal decomposition at temperatures above 140 °C;

Chemical interaction of sulfur with the additive.

Many researchers [92,107,108,109] note that solid sulfur should not be regarded as a homogeneous material but should, to a greater extent, be considered as a composite material, in which part of the allotropes play the role of filler and the other part—as a binder. From the point of view of the theory of strength of composite materials under the action of a load, components with a lower modulus of elasticity (polymer sulfur) are deformed and redistribute the load on the high-modulus component (crystal modifications), which explains the increase in sulfur strength. To obtain an effective sulfur composite, it is sufficient to obtain a material with a partial content of polymer modification. The maximum strength is observed when the content of polymeric sulfur is about 6–8% [50,60,102]. The addition of chemically inert and active ultrafine fillers, as well as organic compounds, is a well-known method [110]. At present, methods based on the addition of nanocarbon and iron-containing modifiers are being actively developed [110,111]. It is worth mentioning here the works showing the efficiency of the introduction of nanocarbon systems (Svatovskaya [112] used iron-containing sols and Magdaleno López et al. [113]—water-soluble iron sols). Prasad et al. [114] investigated the structure formation of polymeric materials in an electromagnetic field. In References [110,111,112,113,114], it was proved that the strength of a polymer composite can be increased by up to 40%. Thus, the search for various modifiers is needed to achieve the characteristics of concrete required for each specific case (Table 5).

#### 5.3.1. The Microstructure of Various Sulfur Modified Composites

The amendment of sulfur can be carried out by polymerization with a cyclic hydrocarbon in compliance with McBee [115,116], by socializing a dicyclopentadiene (DCPD), hydrocarbon polymer compound and molten secondary sulfur in the heat varies in between 120 and 140 °C for 30 min and then rapidly freezing and curing the resulting sulfur polymer.

#### 5.3.2. Melt Viscosity and Mixture Mobility

Liquid sulfur has a dynamic viscosity in the range of (6.5–11) × 10^−3^ Pa⋅s for the temperature range of 120–155 °C (i.e., sulfur is an easily mobile fluid). This allows for the adjustment of the rheology of sulfur materials during heat treatment [77]. The maximum temperature for working with sulfur is 159 °C, then the polymerization of cyclo-octa sulfur begins with the formation of catena-polymers and an increase in viscosity several thousand times to 93.3 Pa·s at 187 °C [121]. The plastic properties of sulfur-based mortars are enhanced by the use of plasticizers (for example, polysulfides); simultaneously with these, the crack resistance of hardening concrete increases. In Reference [118], cast self-compacting sulfur-based concretes were obtained with compressive and flexural strengths of 20 and 12 MPa, respectively. Plasticity in these concrete mixtures was provided by the surface interaction of particles of asphalt granulate with molten sulfur. The workability of the mixture depended on the concentration of sulfur in the binder, the characteristics of fillers and aggregates, the type and modifying additives which significantly affect the structure of sulfur-based concrete.

#### 5.3.3. Hardened Properties

The strength and content of its components, the intensity of physicochemical interactions occurring at the interface, the technology of preparation, molding and so forth, affect the strength of sulfur-based concrete [117]. The strength of sulfur-based concrete decreases with increasing content of aggregate, which is associated with a regular decrease in the content of the most high-strength component of concrete—sulfur binder obtained by combining sulfur, filler and modifying additives [110] (Table 5). Sulfur-based concrete offers rapid attainment of compressive and flexural strength after sulfur binder solidification [18,36,122] (Table 6). In conventional concrete, hydration for 28 days is necessary to attain 90% of the final strength while taking caring about required temperature and moisture conditions. On the other hand, sulfur-based concrete achieves its ultimate strength in few hours without any specific requirements of temperature and moisture [123]. Furthermore, in conventional concrete, the compressive strength increases with increase in strain and eventually decreases after a maximum point (until 0.17 mm/mm and 20 MPa). On the other hand, the compressive strength of sulfur-based concrete also has a linear relation with strain but having higher values (like 0.025 mm/mm and 40 MPa) [122]. X-ray diffraction testing/analysis can be carried out to study the strength development of sulfur-based concrete.

This analysis is used to identify the mineralogical composition which could possibly form during the hardening period [16,25]. The mineralogical matrix of sulfur-based concrete after one day of batching has major constituents like sulfur and silica while minor components are plagioclase, calcite, hematite, dolomite and hydrate of aluminum oxide [124]. High mechanical strength of sulfur-based concrete is justified by the availability of alumino and calcium alumino silicates in their respective oxides like SiO_2_, Al_2_O_3_ and CaO.

#### 5.3.4. Durability Properties

Numerous studies [110,117] have established that the resistance against the acids of sulfur composites in aggressive liquid media depends on the depth of its penetration into the structure of the material. The characteristics of water absorption of sulfuric materials are influenced by a number of factors: the content of sulfur and filler, the type and concentration of modifying additives and so forth. The type and amount of filler and modifying additives also significantly affect the water resistance characteristics of sulfur composites [110,117]. For example, the introduction of paraffin and stearic acid in sulfur composites leads to a slight increase in water resistance and the addition of kerosene, barite and thiokol slightly reduces this figure [49,110]. The resistance of sulfur building materials to acids can also be improved using modifying additives. In particular, in References [88,110,117], it was found that the modification of the sulfur composite with dicyclopentadiene leads to a sharp increase in chemical resistance in salt (0.90–0.98), acid (0.78–0.90) and organic (0.95–0.98) environments. Papers [89,110,117,126,127] proposed to modify the surface of the filler particles with a dressing additive (kerosene solutions of liquid rubbers). The resistance of sulfur-based concrete to various chemical environments and biological agents is given in Table 7, the compositions of which are presented earlier, in Table 6.

#### 5.3.5. Deformative properties

The deformative properties of sulfur-based concrete are taken into account when determining crack resistance and structural rigidity. Deformative characteristics of sulfur-based concrete are given in Table 8 [3,8,11,125]. Some problems arise due to low-temperature creep, which, depending on the formulations and conditions of use of products, may be lower or higher than the creep of ordinary concrete [5].

Since creep is associated primarily with defects in the crystal structure and the presence of extraneous (amorphous) phases, the presence of organic plasticizers in sulfur binder makes a negative contribution to this process [3,110]. Computer simulations of the behavior of plasticized sulfur-based concrete showed that a reduction in creep, along with the greatest possible reduction in the amount of sulfur binder used, can be achieved by compacting the material in a direction towards the progressive hardening front of sulfur, which compensates for shrinkage by the binder movement [10,58]. In other words, the binder must “fill” the contraction of the volume in the transitional state of the system from liquid to solid [11,125]. Sulfur-based concretes have a decaying creep at a level of loading up to 0.5 R_lim_.

Hardened sulfur-based concrete is practically not subject to shrinkage. After 120 days, the signs of the indicators were insignificant and comparable in magnitude to the deformations from temperature fluctuations, that is, proportional to the coefficient of linear thermal expansion [48,125]. The frost resistance of sulfur-based concrete shows a sharp decrease in strength during the first 50 cycles; however, in the future up to 500 cycles, the decrease in strength is quite insignificant.

## 6. Applications of Sulfur in Concrete

In terms of the composition of the components, sulfur-based concrete consists of 70%–90% mineral fillers (aggregates) and 10–30% sulfur binder [3,5,6,7,8,9,10,11,12]. The optimal sulfur content in the material is determined on the basis of the calculated and experimental values of the porosity of compacted mixtures of fillers. Below the optimum content of sulfur, inoperative highly viscous compositions with high porosity and permeability are obtained and above the optimum of sulfur, the adverse effects of volume contraction are manifested—the formation of defects (cracks) and the deformation of crystals with a decrease in strength. It should be borne in mind that the minimum allowable content of sulfur binder is dictated by its function—the matrix, transmitting stress to the grain reinforcing filler (high-modulus component), as well as high cost due to the modifier constituting up to 60% of the cost of sulfur-based concrete. The stages of obtaining sulfur polymer concrete are shown in Figure 10.

Since the typical formation of sulfur-based concrete is the process of impregnation of fillers with liquid plasticized sulfur with its subsequent crystallization during cooling, it is natural to expect an analogy in the behavior of sulfur binder during heating and cooling [45,77]. In general, this is the case. But the adjustments are made by the dispersed phase, the more it, along with the coarsely dispersed component—aggregate (coarse sand, crushed stone) of millimeter size, contains a large proportion of small particles of micron size—filler (fine sand, ash, soot), for which the specific surface area is usually 200–500 m^2^/kg.

## 7. Future Works

Much research has been done to increase the effectiveness of sulfur in concrete production and it has been proven that this waste has useful value for various applications [35,127]. In 2011, Sabour et al. [124] investigated sewage pipes made of sulfur-based concrete. The results showed that sulfur-based concrete was significantly more resistant to strong acid effects (chemical corrosion) compared to cement concrete but less resistant to the effects of microbial corrosion. Sulfur polymer concrete has good potential for the production of concrete blocks [124]. The mixture design, which contains 42% aggregate, 40% natural sand, 11.50% sulfur pellets, 1.2% modified sulfur and 5.3% fly ash, can be used for the manufacture of precast concrete structures. In addition, sulfur is an element distributed on the lunar surface, which can be extracted from lunar soils by heating [10,104,128]. Samples of lunar concrete were prepared that can help design structures from it to minimize the harmful lunar effects [10]. Further research may be directed at eliminating the disadvantages of sulfur-based concrete. The disadvantages of sulfur-based concrete include stringent requirements for the production technology, as well as holding the solution at a temperature of 140 °C [102], therefore thermal stabilizers are necessary [128]. The second disadvantage is that the amount of polymer sulfur in sulfur-based concrete decreases with time, it can turn into a monoclinic form and therefore requires chemical stabilization [107]. The third drawback is the biophilic properties of sulfur. In the presence of moisture and organic matter, some types of bacteria are able to feed on sulfur, for example, sugar [129]. To eliminate such phenomena, microbiological corrosion inhibitors are necessary. In addition, it should be remembered that sulfur is a slightly toxic substance. But it can sublimate toxic substances even in solid form. Therefore, for example, it is necessary to provide for the presence of a special insulating layer in sulfur-based concrete [130,131].

## 8. Conclusions

The development of effective cementless construction materials is relevant for the modern construction industries nowadays. Sulfur is deemed as a cementitious material that can be used as a partial-alternate binder to OPC. Based on previous studies, it has been found that to incorporate sulfur in concrete, sulfur modification is a prerequisite. Moreover, the workability has a significant impact on the structure of sulfur-based concrete that is highly dependent on the concentration of sulfur, the melt viscosity, fillers characteristic, type of aggregates and type and concentration of modifying additives. The characteristics of compressive strength, deformative, bending and tension for various compositions of sulfur-based concrete were extensively discussed. The persistence of sulfur-based concrete in various aggressive environments proved that it is more effective than OPC concrete to counter detrimental effects of severe environments. However, it was reported that sulfur-based concrete was applied as leading construction materials for entire underground utility systems, dams and offshore structures. To this end, this study presented a review on the sources, emissions from construction enterprises and compositions of sulfur; described the production techniques and properties of sulfur; and reviewed related literature to generate comprehensive insights into the potential applications of sulfur in the construction industry. So far, based on this wide-ranging review, the following remarks have been made:

Extensive increase in the use of sulfur production has opened its usability in construction industries worldwide.

The use of sulfur, as a waste material of enterprises, for the production of construction materials is efficient due to waste disposals and contributions to environmental protection.

Sulfur has several melting/freezing points that depend upon the solid allotropes under consideration (melted), temperature and pressure of the mixture.

Different modifiers might be applied to improve different engineering and microstructural properties of sulfur-based concrete.

Sulfur-based concrete achieves its ultimate strength in few hours (3–6 h) without any specific requirements of temperature and moisture, in particular, at room temperature.

The optimal sulfur content in the material is commonly determined based on the theoretical and experimental values of the porosity of compacted mixtures of fillers.

Sulfur-based concrete is more resistant to strong acid effects compared to OPC-based concrete.

Incorporation of sulfur in concrete supports sustainability by reducing sulfur emissions through different industries (by capturing) as well as decreasing cement production.

## Figures and Tables

**Figure 1 materials-13-04712-f001:**
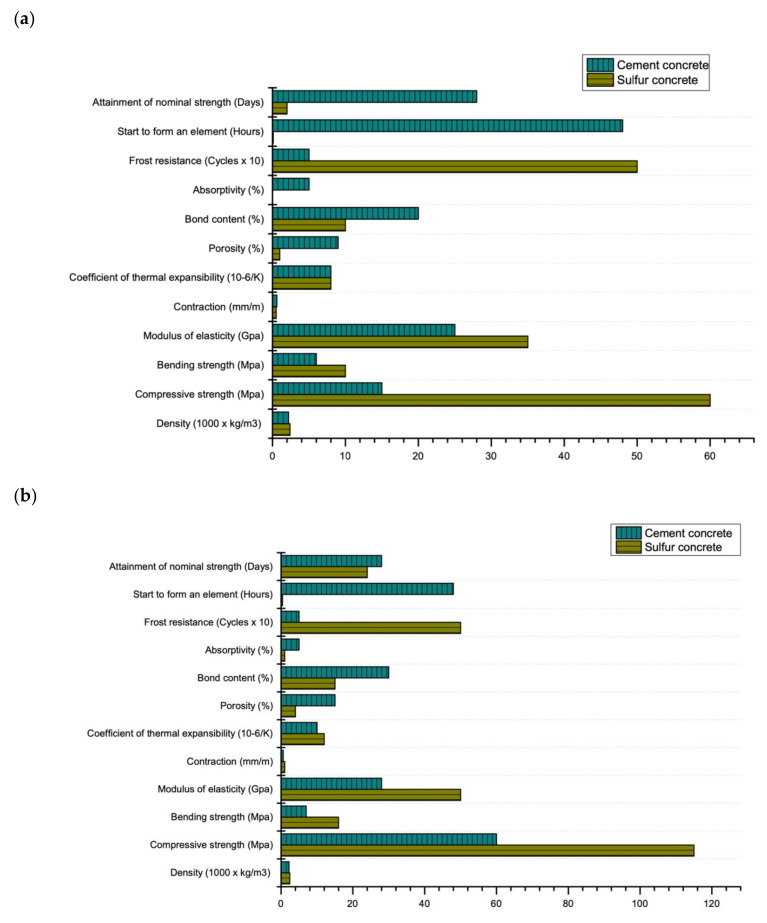
Mechanical and durability properties of cement and sulfur-based concrete: (**a**) maximum (grade M60) (**b**) minimum (grade M15) (data extracted from Reference [14]).

**Figure 2 materials-13-04712-f002:**
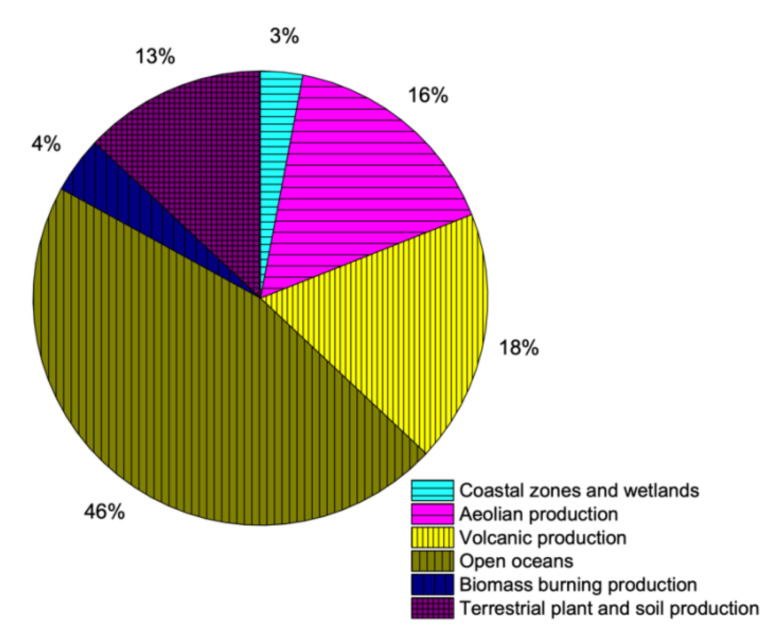
Contribution of sulfur through natural resources (data extracted from Reference [36]).

**Figure 3 materials-13-04712-f003:**
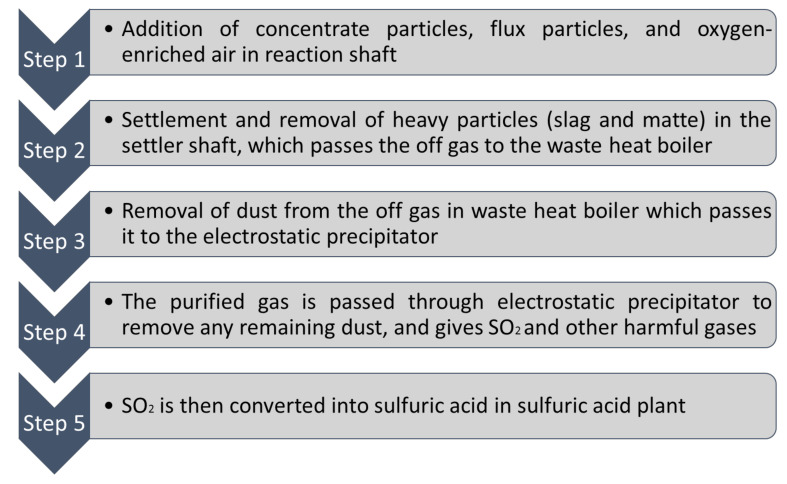
Schematic of Flash Smelting Furnace (data extracted from Reference [46]).

**Figure 4 materials-13-04712-f004:**
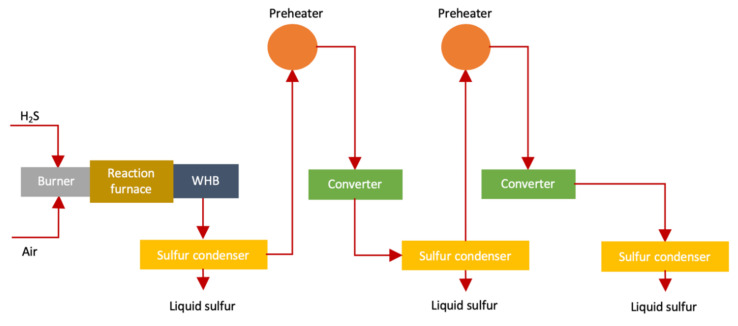
Claus process for sulfur production (adapted from Reference [64]).

**Figure 5 materials-13-04712-f005:**
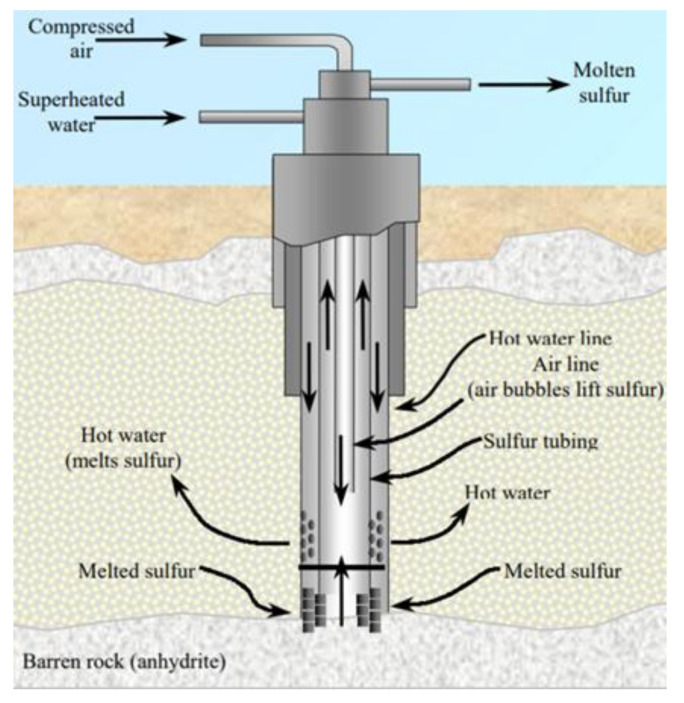
Schematic shape of Frasch pump (courtesy of USGS [1]).

**Figure 6 materials-13-04712-f006:**
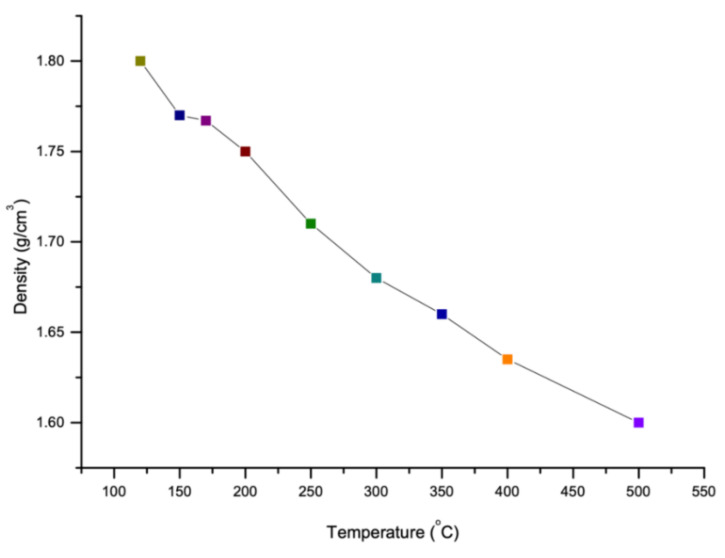
Density of sulfur in accordance with temperature and allotropes (data extracted from References [15,36,69,79,80,81,82,83,84,85,86].

**Figure 7 materials-13-04712-f007:**
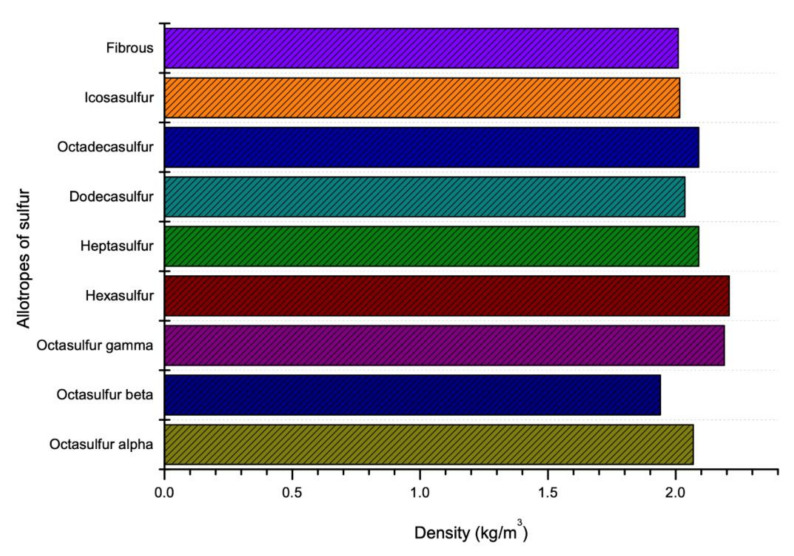
Density of sulfur in accordance with temperature and allotropes (data extracted from References [18,49,57,59,60,61,62,63,64,65,66]).

**Figure 8 materials-13-04712-f008:**
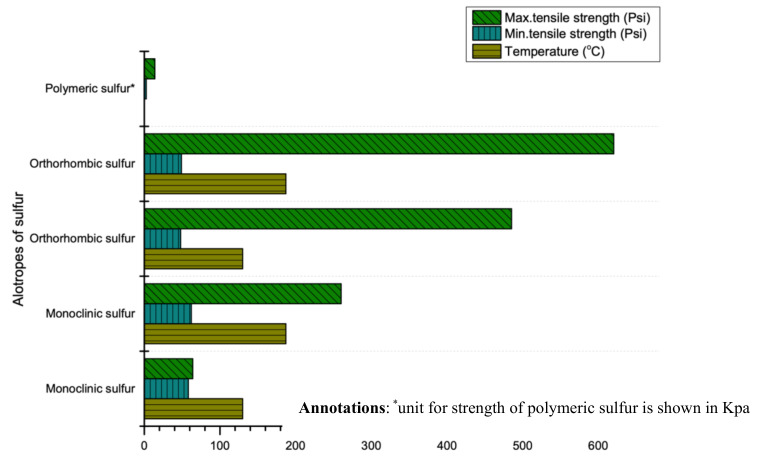
Tensile strength range of sulfur at various temperatures (data extracted from Reference [25]).

**Figure 9 materials-13-04712-f009:**
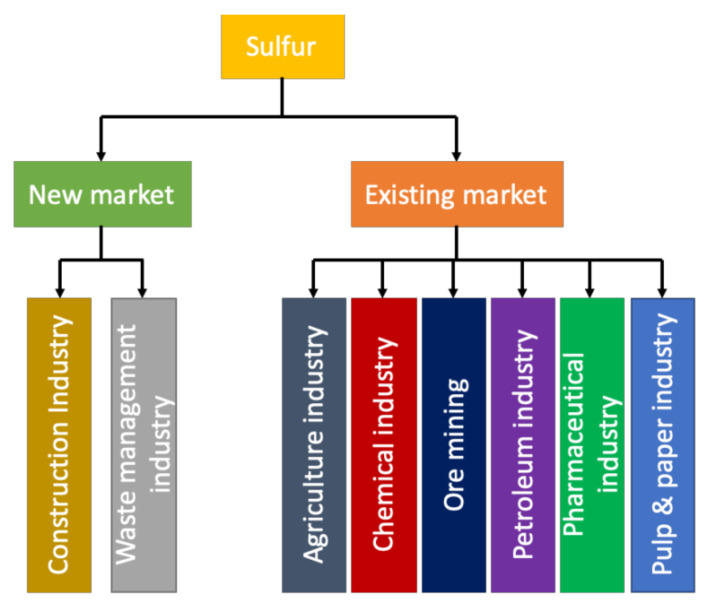
New and existing market of sulfur worldwide (data extracted from Reference [36]).

**Figure 10 materials-13-04712-f010:**
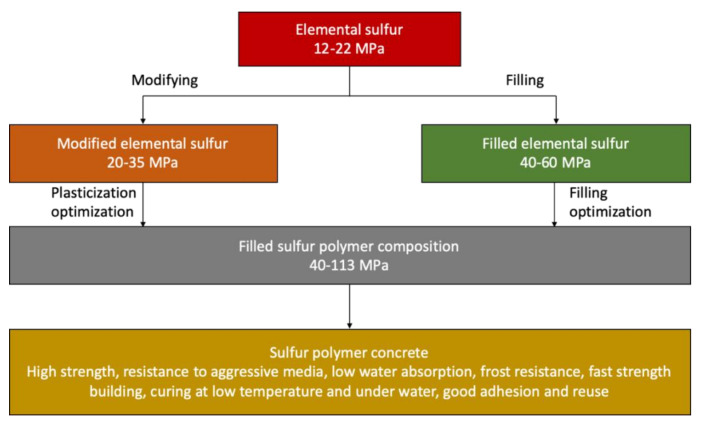
The stages of obtaining sulfur polymer concrete (data extracted from References [3,5,6,7,8,9,10,11,12]).

**Table 1 materials-13-04712-t001:** Comparison between Ordinary Portland Cement (OPC)-based concrete and sulfur-based concrete properties (grade M15 and M60).

Property	Unit	OPC-Based Concrete	Sulfur-Based Concrete	Refs
Time of strength gain	Time	28 days	3 h	[7]
Compressive strength	MPa	15–25	55–65	[14]
Tensile strength	MPa	3–4	5–7	[25]
Wearing capacity	%	17	3	[26]
Flexural strength	MPa	6–9	10–15	
Freezing resistance	%	50	300	[4]
Acids resistance	at 100/cent humidity	23	84	[27]
Water resistance	%	0.8	1.0	[28]

**Table 2 materials-13-04712-t002:** Melting point of various allotropes of sulfur.

Allotrope of Sulfur	Melting Point (°C)	Refs.
α-sulfur	110.06	[36]
	115.1	[13]
	112.8	[36,69]
β-sulfur	114.6	[69]
	119.6	[71]
	120.4	[13]
	133	[69]
γ-sulfur	106.8	[72]
	108	[73]
	108.6	[13]
δ-sulfur	160	[13]
ω-sulfur	77	[73]
	90	
	160	
	104	[69]
Fibrous	75	[73]
	104	[72]
Hexasulfur	50	[74]
Heptasulfur	39	[75]
Cyclo-S_12_	148	[74]
Cyclo-S_18_	128	[76]
Cyclo-S_20_	124	

**Table 3 materials-13-04712-t003:** Colors of various allotropes of sulfur.

Allotrope of Sulfur	Color	Refs
Octasulfur alpha	Bright yellow	[15,76,80,81,82,83,84,85,86]
Octasulfur beta	Yellow	
Octasulfur gamma	Light yellow	
Hexasulfur	Orange to red	
Heptasulfur	Light yellow	
Anneasulfur	Deep yellow	
Decasulfur	Yellow to green	
Octadecasulfur	Lemon to yellow	

**Table 4 materials-13-04712-t004:** Comparison of sulfur and cement concrete (↑: High, ↓: Low and =: equivalent) [14,95].

Properties of Concrete	Sulfur-Based Concrete Compared to Cement Concrete
Wear resistance	↑
Permeability	↓
Bond strength to concrete/reinforcing steel	↑
Thermal conductivity	↓
Elastic modulus	↑
Flexural strength	↑
Compressive/flexural/tensile strength	↑
Fire resistance	↓
Durability during thermal cycles	= or ↑
Corrosion resistance	↑
Fatigue resistance	↑
Linear expansion coefficient	=
Compression creep	↓

**Table 5 materials-13-04712-t005:** Modifiers and fillers for sulfur-based concrete.

Inorganic Additives		
Modifier	Concentration % of the Mass of Sulfur	Result
Talk [3,110]	26	Acid resistance
Heavy metals and mercury [36]	6	Durability
Alumina [3]	20–26	Acid resistance
Fly ash [3]	22–23	
Silica [3]	22–25	
Organic additives		
Dicyclopentadiene [5,9,12,117,118]	0.1–50	Increased strength in corrosive chemical environments, increased fire resistance
Dicyclopentadiene + cyclopentadiene + dipentene [5,9,12]	1–30	Rapid development of compressive strength
Olefin polysulfide additives [12,110]	5–25	Improving the strength characteristics
Epoxy resin [119]	2–6	Improving the strength characteristics
Polyolefin [7]	2.5–5	Plasticizer
Bitumen [6,12,110]	1–4	High corrosion resistance, high physical strength
Additive STX (Starcrete) [120]	2–7	High fatigue strength
Styrene [6,110]	2–30	Low water permeability
Ethylidene norbornene [121]	1–5	Provides, with smaller quantities of the modifier, an increase in the resistance of sulfur-based concrete in acidic and basic environments, has high strength, high frost resistance and also eliminates the toxicity of the material obtained

**Table 6 materials-13-04712-t006:** Characteristics of strength of various sulfur-based concretes.

Authors	Content	Compressive Strength, MPa	Flexural Strength, MPa	Tensile Strength, MPa
Vlahovich et. al. [3]	Sulfur—30 wt. % Sand—63 wt. % Fillers—7 wt. %	55	8	3
Dehestani et. al. [6]	Sulfur—98 wt. % styrene—2 wt. %	54	8	3
Al-Otaibi et al. [7]	modified sulfur—1 wt. % granulated sulfur—11 wt. % sand—40 wt. % coarse aggregate, 42 wt. % fly ash—6 wt. %	30	2	1
Gracia et. al. [50]	Sulfur—25 wt. % Sand—70 wt. % slag—5 wt. %	70	12	5
Bae et. al. [88]	modified sulfur—15 wt. % fly ash—13 wt. % sand—32 wt. % coarse aggregate—40 wt. %	83	13	6
Choura et. al. [92]	Sulfur—50 wt. % Phosphogypsum—50 wt. %	41	5	2
Gwon et al. [118]	modified sulfur—40 vol. % sand—35 vol. % binary cement—25 vol. %	62	9	4
Anyszka et. al. [119]	modified sulfur—30 wt. % sand—70 wt. %	115	16	7
Lopez et. al. [120]	Sulfur—17 wt. % Polymer—2 wt. % Sand—49 wt. % coarse aggregate—24 wt. % soil—8 wt. %	60	13	6
Dugarte et. al. [123]	modified sulfur—30 vol. % sand—70 vol. %	43	5	2
Sabour et. al. [124]	Sulfur—25 wt. % Sand—70 wt. % slag—5 wt. %	52	8	3
Yeoh et. al. [125]	Sulfur—1 wt. % Cement—14 wt. %S and—30 wt. % coarse aggregate—40 wt. % water—15 wt. %	42	5	2

**Table 7 materials-13-04712-t007:** Resistance of sulfur-based concretes.

Author	Lost Weight, %					
	H_2_SO_4_	HCl	NaCl	SO_4_(NH_4_)_2_	Kerosene	Thiobacillus Thiooxidans Bacterium
Sabour et. al. [124]	5	1	-	-	-	2.25
Gwon et al. [117]		0	-	1	-	-
Vlahovich et. al. [3]	0	−1	2	-	-	-
Dehestani et. al. [6]	-	0	-	-	3	-
Dugarte et. al. [123]	1	−1	3	1	-	-
Gracia et. al. [50]	-	-	-	-	3	-
Yeoh et. al. [125]	4	-	2	-	-	-
Choura et. al. [92]	2	-	-	-	-	-
Bae et. al. [88]	3	-	3	1	-	-
Anyszka et. al. [119]	2	-	-	-	-	-
Lopez et. al. [120]	5	-	2	-	3	-
Al-Otaibi et al. [7]	4	-	-	1	-	-

**Table 8 materials-13-04712-t008:** Deformative properties of sulfur-based concretes.

Properties	Concrete on Sulfur and Aggregates		Refs
	Dense	Porous	
Poisson’s ratio	0.19–0.21	0.24–0.31	[3,8,11,125]
Thermal expansion coefficient, 10^−6^ °C^−1^	9–14	7–9	
Linear shrinkage, %	0.9–1.5	0.7–1.1	

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
