# Peer review of "A Critical Review on the Properties and Applications of Sulfur-Based Concrete"

_materials, 2020, doi:10.3390/ma13214712_

Round 1
Reviewer 1 Report
This paper presents an interesting approach on a promising material. After careful evaluation, I would suggest the following:
- ln 23 - the acronym OPC should be detailed as ordinary Portland cement (as in line 122).
- The terms for the parameters in table 1 should be reformulated in proper English (e.g. what does cold resistance mean?; flexing = flexural etc.).
- ln 231 - in my opinion, there should be an explicit chemical reaction.
- lns 252-257, 366-379, 400-402, 578-592 should be bullet lists.
- The sentence on ln 326-328 should be reformulated.
- The two reasons on lns 353-355 should be inserted in the previous phrase.
- ln 512 - optimum content of sulfur.
- The font size is not uniform throughout the text.
- The phrase on lns 510-512 is very general. Details on how the optimal sulfur content is determined should be provided.
- The disadvantages indicated on lns 550-560 should be given a separate, more detailed paragraph.
- ln 570 the syntagm "superior advantages" is a pleonasm.
- Given all its advantages and the fact that it has been developed since the 1970s, why hasn't this material been used much more extensively in the construction industry? Perhaps a discussion on this topic would be useful.
Author Response
Dear Reviewer. Thank you so much for your comments that helped make our manuscript even better! The attached file contains the responses to the comments of all four reviewers!

Reviewer 2 Report
The critical review on the use of sulfur-based concrete is carried out by the authors with a remarkable precision of detail. All potentially interested research fields have been analyzed, ranging from production techniques to the comparison of the main mechanical parameters of strength and durability compared to traditional concrete. The paper is certainly worthy of publication. It is only required to better specify the abbreviation OPC in the text and to review the temporal sequence of the citations within the text, as it seems that the numbering is not consecutive to the sequential appearance of the citation.
Author Response

(The authors gave the same response as above.)

Reviewer 3 Report
The authors present indeed comprehensive review on the characterization and usage of sulfur in civil engineering as an alternative binder for concrete production. It is clear that they are familiar with that area. Although the manuscript is properly done, I met some shortcoming to be improved, find them as follows:
- The text itself is well written and easy to follow, however, some parts are a bit tacky. See for example lines 26 and 27 where commas are missing, two individual sentences are nor separated, etc. Therefore, the text has to be revised again to avoid these shortcomings.
- The statement in lines 83–84 is too vague. It should mention more specific information.
- References should be sorted in ascending order through the text, see for example reference nr. 65 in line 99 and several following.
- Figure 1 summarizes cement and sulfur concrete properties. It should be added, which concrete grade/type was considered as reference material (for example C12/15 and C60/75 according to European technical standards or other alternatives). This holds also for Table 1 where basic parameters of standard concrete are listed.
- Table 1: use standard terminology when describing mechanical properties of concrete (compressive strength, tensile strength, flexural or bending strength).
- Figure 7 shows density as a function of temperature starting with a very high temperature value (about 125 °C), missing fewer temperatures. Why?
- Chapter 4.6 Strength (I suggest to rename it as Mechanical properties) has to be extended. Figure 9 shows only tensile strength, while other parameters like compressive and flexural strength are also important and deserve a graphical representation as well.
- Table 6, column 2: specify whether you used wt. or vol. %.
Author Response

(The authors gave the same response as above.)

Reviewer 4 Report
The paper topic is very interesting and the paper is very well written. There are few editorial comments:
Is it necessary to put table of contents into the paper?
Maybe would be better to show parameters on figure 1 into the same diagram, not separately minimum and maximum values.
In table one, one of the parameters is flexing strength - why not flexural strength - maybe this is typing error.
Maybe I didnt understand something but why is production of sulphor describbed when the intention of usage of suphor in concrete is to reduce sulphor waste in the world. Why should we produce the sulphor as we have it as waste of some other industrial processes...
Author Response

(The authors gave the same response as above.)
